# Investigation of the Sn-0.7 wt.% Cu Solder Reacting with C194, Alloy 25, and C1990 HP Substrates

Andromeda Dwi Laksono [ID], Tzu-Yang Tsai, Tai-Hsuan Chung, Yong-Chi Chang and Yee-Wen Yen *[ID]

Department of Materials Science and Engineering, National Taiwan University of Science and Technology, Taipei 10672, Taiwan
* Correspondence: ywyen@mail.ntust.edu.tw

**Abstract:** Cu-based alloys are one of the most promising substrates to enhance the performance of lead-frame materials. In the present study, the interfacial reactions in the Sn-0.7 wt.% Cu (SC) lead-free solder reacting with Cu-3.3 wt.% Fe (C194), Cu-2.0 wt.% Be (Alloy 25), and Cu-3.3 wt.% Ti (C1990 HP) were investigated. The material underwent a liquid–solid interface reaction, and the reaction time was 0.5 to a few hours at the reaction temperatures of 240 °C, 255 °C, and 270 °C. The morphology, composition, growth rate, and growth mechanism of the intermetallic compounds (IMCs) formed at the interface were investigated in this study. The results showed that the reaction couples of SC/C194, SC/Alloy 25, and SC/C1990 HP formed IMCs, which were the [$(Cu, Fe)_6Sn_5$ and $(Cu, Fe)_3Sn$], [$(Cu, Be)_3Sn$ and $(Cu, Be)_6Sn_5$], and [$Cu_6Sn_5$] phases, respectively. Finally, the IMC growth mechanism for the SC/C194, SC/Alloy 25, and SC/C1990 HP couples displayed reaction control, grain boundary diffusion control, and diffusion control, respectively.

**Keywords:** interfacial reaction; Sn-0.7 wt.% Cu (SC) lead-free solder; C194; Alloy 25; C1990 HP





## 1. Introduction

In this past, a tin–lead solder was the most common solder, with excellent soldering properties and reliability [1–5]. However, lead is reported to be a toxic metal that will affect the human body and cause environmental pollution. Therefore, the traditional tin–lead solders have been replaced with lead-free solders. Nowadays, the Sn-0.7 wt.% Cu (SC) solder is one of the most popular lead-free solders. As a result of its low cost and resistance to oxidation, the SC solder is widely used in the electronic industry [6–9]. Cu is widely used in lead-frame materials and metal substrates, owing to its good electrical conductivity and mechanical properties. Some trace elements can be added to the Cu-based substrate to form an excellent alloy. The formation of a copper–iron alloy (C194) has been reported via the addition of a 3.3 wt.% Fe element to Cu, which is known for its high strength, fine crystal structure, strong corrosion resistance, and good weldability [10]. Meanwhile, the addition of 2.0 wt.% Be element to Cu produced a beryllium–copper alloy (Alloy 25) with enhanced electrical conductivity, mechanical properties, weldability, and fatigue resistance, which is also commonly used in the electronic industry [11]. Similarly, the addition of 3.3 wt.% Ti to Cu produced a copper–titanium alloy (C1990 HP) with good strength, electrical conductivity, workability, fatigue, and corrosion resistance [12]. These substrates have the potential to be suitable as lead-frame materials to change the conductor Cu, since many researchers have conducted experiments related to their reliability [11,13–18].

However, the reaction of a tin-based solder with these three different alloys has been less discussed. Thus, to explore the type and morphology of the IMCs formed at the interface, the present study investigates the interfacial reactions between the SC solder and C194, Alloy 25, and C1990HP substrates as liquid–solid reaction couples at the reaction temperatures of 240 °C, 255 °C, and 270 °C for reaction times ranging from 0.5 h to 10 h. Moreover, the IMC growth mechanism and kinetics were elucidated according to the

thickness of the IMCs formed at different reaction temperatures and times. The present study can assist the electronic packaging industry by revealing the applicability of C194, Alloy 25, and C1990HP.

## 2. Materials and Methods

The compositions of C194, Alloy 25, and C1990 HP substrates were determined with an electron probe micro-analyzer (EPMA Joel, JXA-8200; Tokyo, Japan) using wavelength dispersive spectrometers (WDS), as shown in Table 1. The C194, Alloy 25, and C1990 HP substrates were cut into dimensions of $10 \times 7 \times 0.2$ mm$^3$ and polished to achieve a bright and scratch-free surface. An ultrasonic oscillator was used to ensure the cleanliness of the substrate surface. After cleaning for 5 min in the ultrasonic oscillator, acetone was used to remove oil, and hydrochloric acid was used to remove oxides. Finally, the substrate surface was cleaned with alcohol and deionized water, followed by drying.

**Table 1.** Chemical composition of C194, Alloy 25, and C1990 HP (in wt.%) substrates.

| Components | C194 | Alloy 25 | C1990 HP |
|---|---|---|---|
| Cu | 97.8 | 97.8 | 96.7 |
| Fe | 2.2 | - | - |
| Be | - | 2.2 | - |
| Ti | - | - | 3.3 |

For the solder preparation, a commercial SC solder (Fuzhou Weiju Trading Co., Ltd., Fuzhou, China) was used. The weight of the SC solder was three times the weight of the substrate. In addition, to allow the melting solder to be attached to the substrate, first, a rosin mildly activated (RMA) flux was applied on both sides of the substrate, and then the substrate and SC were placed perpendicularly to a quartz tube glass. After this, a high-temperature oxygen-gas flame was used to seal the quartz tube. Finally, the reaction couple was placed into a furnace for liquid–solid interfacial reaction at the reaction temperatures of 240 °C, 255 °C, and 270 °C, respectively, for reaction times ranging from 0.5 h to 10 h. When the reaction was finished, the reaction couple was taken out of the furnace and immediately quenched in ice water. Subsequently, the reaction couple could be taken out by breaking the quartz tube. The reaction couples, SC/C194, SC/Alloy 25, and SC/C1990HP, were hot-mounted with conductive bakelite powder. The surface of the mounted sample was finely ground with 120-, 400-, 800-, 1200-, 2500-, and 4000-grit SiC sandpapers. After polishing with 1.0 μm and 0.3 μm alumina powder polishing solutions, the metallographic treatment was conducted for the embedded samples. The microstructure and growth morphology were studied using an optical microscope (OM; Olympus BX51M, Tokyo, Japan) and scanning electron microscope (SEM; TM-3000, Tokyo, Japan). Energy-dispersive spectrometry (EDS; Bruker, Quantax 70; Berlin, Germany) was performed to analyze the composition of the IMCs formed at the interface. The results were compared with a phase diagram and related literature. Finally, the samples were quantitatively analyzed using EPMA to ensure the composition of each position. To ensure and explain the mechanism of the IMC phenomena in the SC/Alloy 25 system, an atomic force microscope (AFM) and focused ion beam (FIB) microscope were used. For the quantitative analysis of each diffusion layer structure, the average thickness of the intermetallic phase was measured using image analysis software (Image J). To investigate the in-depth morphology of the IMC, the samples were immersed in an etching solution. The composition of the etching solution used was $CH_3OH:HCl:HNO_3 = 93:2:5$ [19].

## 3. Results

### 3.1. SC/C194 Reaction Couples

The backscattered electron image (BEI) micrographs of SC and C194 alloy at reaction temperatures ranging from 240 °C to 270 °C for different reaction times are shown in Figure 1a–i. The presence of the C194 substrate is demonstrated by the dark part of the

bottom layer, and the presence of the SC solder is demonstrated by the bright part. When the reaction starts, the metal atoms diffuse, and the IMC will form at the interface [20]. Figure 1a shows the BEI micrograph of the IMC formed via interfacial reactions between the SC solder and C194 at 240 °C after 0.5 h of reaction time. It was confirmed by EDS that the top layer of the IMC was composed of Cu-45.6 at.% Sn-3.7 at.% Fe and was determined as the $Cu_6Sn_5$ phase by comparison with the Cu-Sn binary phase diagram [21]. The phase of IMC was denoted as the $(Cu, Fe)_6Sn_5$ phase because of the Fe concentration inside the IMC. Another extremely thin IMC layer was observed near the end of the C194, which was composed of Cu-25.6 at.% Sn-5.0 at.% Fe, determined as the $(Cu, Fe)_3Sn$ phase. The above results are consistent with the study of Xie et al. [22] on SC/Cu, which reported the existence of $Cu_6Sn_5$ and $Cu_3Sn$ at the interface after the reaction between the Cu substrate and SC solder. Figure 1b illustrates the BEI micrograph of the IMC formed via interfacial reactions between the SC solder and C194 at 240 °C after a 4 h reaction time. It was found that the grains of $(Cu, Fe)_6Sn_5$ had ripened and started to disperse due to the presence of a small amount of Fe atoms, which became nucleation points prone to heterogeneous. The above results resemble those of the liquid–solid interface reaction of SC/Cu studied by Lai et al. [23], but no scallop-shaped $(Cu, Fe)_6Sn_5$ phase was formed. Figure 1c shows the BEI micrograph of the IMC formed via interfacial reactions between the SC solder and C194 at 240 °C after 10 h of reaction time. As shown in Figure 1c, only a very thin $(Cu, Fe)_3Sn$ phase was observed. Moreover, the thickness of $(Cu, Fe)_3Sn$ had no obvious growth, and no scalloped formation of $(Cu, Fe)_6Sn_5$ was observed, which might be due to the occurrence of heterogeneous nucleation, and the Cu atomic concentration cannot be accumulated at the interface. These results are consistent with a reported study on the liquid–solid interface reaction between Sn3.5Ag0.5Cu (SAC) and Fe, which showed that the addition of Fe atoms reduces the grain size and inhibits the growth of the $Cu_3Sn$ phase [24]. Similarly, it has been reported that the spalling phenomenon of $Cu_6Sn_5$ becomes more intense with an increase in the reaction time [25]. The phase formed and phenomena observed in the SC/C194 couple at a 240 °C reaction temperature also occurred at 255 °C and 270 °C reaction temperatures, as shown in Figure 1d–i and Table 2. To study the in-depth morphology of the IMC, the SC solder above the IMC was completely removed by etching, as shown in Figure 2. The BEI micrograph of the shallow etched SC/C194 interfaces reacted at 240 °C, 255 °C, and 270 °C reaction temperatures for 0.5 h and 2 h each is shown in Figure 2a–g. According to Figure 2a,c,e, all the $Cu_6Sn_5$ phases formed were small and long columns. Meanwhile, the $Cu_6Sn_5$ phase was also gradually transformed from the original small columnar structure to a larger hexagonal columnar body as the reaction time was extended to 2 h, and the IMCs were stacked loosely, as shown in Figure 2b,d,f. According to Figure 2b,d,f, the IMCs became more dispersed in the form of small islands with the increase in the reaction temperature. The typical hexagonal type of the $(Cu, Fe)_6Sn_5$ (Figure 2g) phase is consistent with the literature [26]. Moreover, it has been reported that the morphology of $Cu_6Sn_5$ in the SC/Cu system has a scallop-like structure [27]. The Cu atoms are accumulated at the interface to form a thicker $Cu_3Sn$ phase, and the dispersion phenomenon of the IMC becomes more significant with the increase in the reaction temperature and time [28].

**Table 2.** Evolution of IMCs in the liquid–solid reactions in various couples at various temperatures and reaction times.

| | SC/C194 Couples | | | | | |
|---|---|---|---|---|---|---|
| | **Reaction Time (h)** | | | | | |
| **Temp (°C)** | **0.5** | **1** | **2** | **4** | **5** | **10** |
| 240 | $(Cu, Fe)_6Sn_5$ $(Cu, Fe)_3Sn$ | $(Cu, Fe)_6Sn_5$ $(Cu, Fe)_3Sn$ | $(Cu, Fe)_6Sn_5$ $(Cu, Fe)_3Sn$ | $(Cu, Fe)_6Sn_5$ $(Cu, Fe)_3Sn$ | $(Cu, Fe)_6Sn_5$ $(Cu, Fe)_3Sn$ | $(Cu, Fe)_6Sn_5$ $(Cu, Fe)_3Sn$ |
| 255 | $(Cu, Fe)_6Sn_5$ $(Cu, Fe)_3Sn$ | $(Cu, Fe)_6Sn_5$ $(Cu, Fe)_3Sn$ | $(Cu, Fe)_6Sn_5$ $(Cu, Fe)_3Sn$ | $(Cu, Fe)_6Sn_5$ $(Cu, Fe)_3Sn$ | $(Cu, Fe)_6Sn_5$ $(Cu, Fe)_3Sn$ | $(Cu, Fe)_6Sn_5$ $(Cu, Fe)_3Sn$ |

**Table 2.** *Cont.*

| SC/C194 Couples | | | | | |
|---|---|---|---|---|---|
| **Reaction Time (h)** | | | | | |

| Temp (°C) | 0.5 | 1 | 2 | 4 | 5 | 10 |
|---|---|---|---|---|---|---|
| 270 | $(Cu, Fe)_6Sn_5$ $(Cu, Fe)_3Sn$ | $(Cu, Fe)_6Sn_5$ $(Cu, Fe)_3Sn$ | $(Cu, Fe)_6Sn_5$ $(Cu, Fe)_3Sn$ | $(Cu, Fe)_6Sn_5$ $(Cu, Fe)_3Sn$ | $(Cu, Fe)_6Sn_5$ $(Cu, Fe)_3Sn$ | $(Cu, Fe)_6Sn_5$ $(Cu, Fe)_3Sn$ |

| SC/Alloy 25 Couples | | | | | | |
|---|---|---|---|---|---|---|
| **Reaction Time (h)** | | | | | | |

| Temp (°C) | 0.5 | 1 | 2 | 4 | 5 | 10 |
|---|---|---|---|---|---|---|
| 240 | $(Cu, Be)_6Sn_5$ | $(Cu, Be)_6Sn_5$ $(Cu, Be)_3Sn$ | $(Cu, Be)_6Sn_5$ $(Cu, Be)_3Sn$ | $(Cu, Be)_6Sn_5$ $(Cu, Be)_3Sn$ | $(Cu, Be)_6Sn_5$ $(Cu, Be)_3Sn$ | $(Cu, Be)_6Sn_5$ $(Cu, Be)_3Sn$ |
| 255 | $(Cu, Be)_6Sn_5$ $(Cu, Be)_3Sn$ | $(Cu, Be)_6Sn_5$ $(Cu, Be)_3Sn$ | $(Cu, Be)_6Sn_5$ $(Cu, Be)_3Sn$ | $(Cu, Be)_6Sn_5$ $(Cu, Be)_3Sn$ | $(Cu, Be)_6Sn_5$ $(Cu, Be)_3Sn$ | $(Cu, Be)_6Sn_5$ $(Cu, Be)_3Sn$ |
| 270 | $(Cu, Be)_6Sn_5$ $(Cu, Be)_3Sn$ | $(Cu, Be)_6Sn_5$ $(Cu, Be)_3Sn$ | $(Cu, Be)_6Sn_5$ $(Cu, Be)_3Sn$ | $(Cu, Be)_6Sn_5$ $(Cu, Be)_3Sn$ | $(Cu, Be)_6Sn_5$ $(Cu, Be)_3Sn$ | $(Cu, Be)_6Sn_5$ $(Cu, Be)_3Sn$ |

| SC/C1990HP Couples | | | | | |
|---|---|---|---|---|---|
| **Reaction Time (h)** | | | | | |

| Temp (°C) | 0.5 | 1 | 2 | 4 | 5 |
|---|---|---|---|---|---|
| 240 | $Cu_6Sn_5$ | $Cu_6Sn_5$ | $Cu_6Sn_5$ | $Cu_6Sn_5$ | $Cu_6Sn_5$ |
| 255 | $Cu_6Sn_5$ | $Cu_6Sn_5$ | $Cu_6Sn_5$ | $Cu_6Sn_5$ | $Cu_6Sn_5$ |
| 270 | $Cu_6Sn_5$ | $Cu_6Sn_5$ | $Cu_6Sn_5$ | $Cu_6Sn_5$ | $Cu_6Sn_5$ |

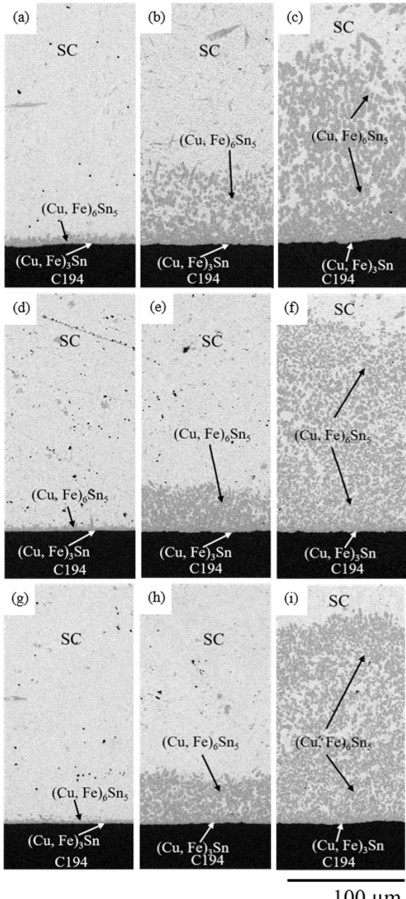

**Figure 1.** BEI micrographs of the SC/C194 reaction couples at 240 °C for (**a**) 0.5 h, (**b**) 4 h, and (**c**) 10 h; reacted at 255 °C for (**d**) 0.5 h, (**e**) 4 h, and (**f**) 10 h; and reacted at 270 °C for (**g**) 0.5 h, (**h**) 4 h, and (**i**) 10 h.

To understand the atomic diffusion behavior, verify the formation of IMCs, and observe the dispersion phenomenon, EPMA was conducted. Figure 3 shows the BEI micrograph and EPMA line-scan analysis of the SC/C194 reaction couple reacted at a 240 °C reaction temperature for a 5 h reaction time. The composition and distance relationship were elucidated through an EPMA line scan of the SC/C194 reaction couple reacted at a 240 °C reaction temperature for a 5 h reaction time. It can be speculated that the SC started to diffuse from the solder side to the C194 side, which resulted in the formation of the IMC. The first layer of the IMC was $(Cu, Fe)_3Sn$, which is Cu-25.1 at.% Sn-2.3 at.% Fe, and the layer on top was $(Cu, Fe)_6Sn_5$. According to a previous report, the Cu content of the $Cu_6Sn_5$ phase ranges from 52.2 at.% to 54.5 at.% [29]. The measured composition was consistent with the literature and EDS analysis results.

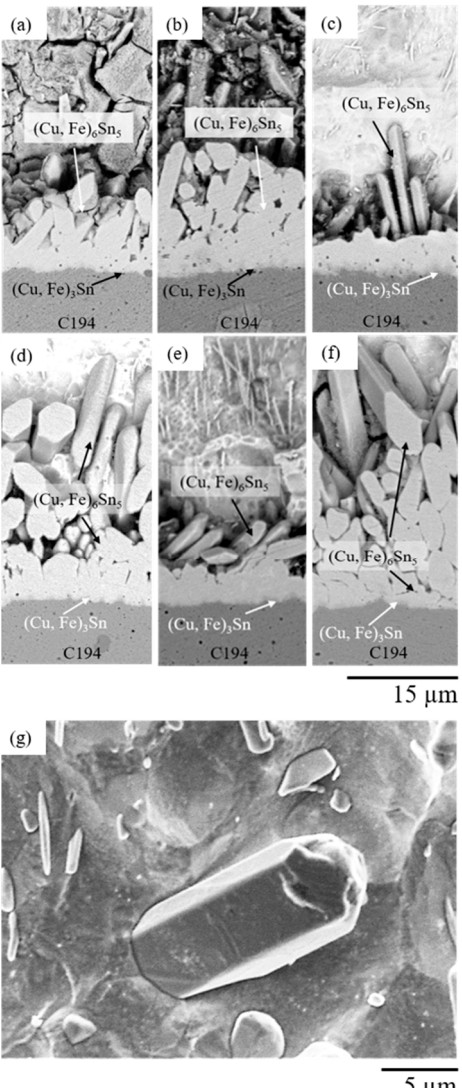

**Figure 2.** BEI micrographs of the shallow etched SC/C194 interfaces reacted at 240 °C for (**a**) 0.5 h and (**b**) 2 h; reacted at 255 °C for (**c**) 0.5 h and (**d**) 2 h; and reacted at 270 °C for (**e**) 0.5 h and (**f**) 2 h; (**g**) BEI partial enlarged micrograph of the SC/C194 couples.

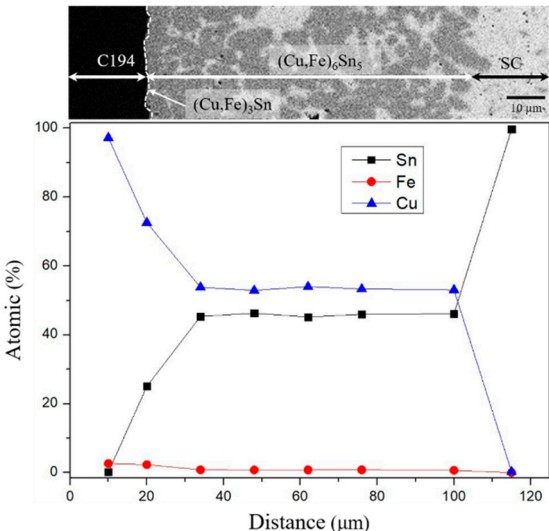

**Figure 3.** EPMA line scan and corresponding BEI micrograph of SC/C194 reaction couple at a reaction temperature of 240 °C and reaction time of 5 h.

### 3.2. SC/Alloy 25 Reaction Couples

The BEI micrograph of the SC/Alloy 25 reaction couple reacted at a 240 °C reaction temperature for a 0.5 h reaction time is shown in Figure 4a. It was found that a dense layered IMC was formed on the substrate side, with a composition of Cu-46.6 at.% Sn-0.8 at.% Be, denoted as $(Cu, Be)_6Sn_5$. It reflected the incorporation of a small amount of Be into the Cu sublattice of the $Cu_6Sn_5$ phase, as shown in the previous literature [15]. The BEI micrograph of the reaction couple of SC/Alloy 25 reacted at a 240 °C reaction temperature for a 1 h reaction time is shown in Figure 4b. As the reaction time reached 1 h, a very thin layer with composition Cu-25.4 at.% Sn-4.9 at.% Be was formed, as shown in Figure 4b. According to the Sn-Cu phase diagram [21], the IMC was determined to be the $(Cu, Be)_3Sn$ phase. However, the $(Cu, Be)_3Sn$ phase had not been observed for 0.5 h owing to the insufficient reaction time. Furthermore, the Cu atomic concentration at the interface could not accumulate to form the Cu-rich $(Cu, Be)_3Sn$ phase. The results are consistent with a previous study of Sn/Alloy 25 reaction couples [15]. The EPMA result showed that the composition of the upper layer was Cu-46.0 at.% Sn, and that of the lower layer was Cu-42.9 at.% Sn, as shown in Figure 5. These two layers should be the $(Cu, Be)_6Sn_5$ phase compared with a binary phase diagram. The BEI micrograph of the reaction couple of SC/Alloy 25 reacted at a 240 °C reaction temperature for a 4 h reaction time is shown in Figure 4c. It can be observed that the $(Cu, Be)_6Sn_5$ appeared to be divided into upper and lower layers as the reaction time reached 4 h, as shown in Figure 4c.

The temperature increases up to 255 °C for a 0.5 h reaction time on the SC/Alloy 25 couple could be observed from the BEI micrograph, as shown in Figure 4e. It was observed that the IMC with a scallop-like structure appeared on the solder side, which was the same as the result obtained at 240 °C. According to EDS, the composition of the IMC was Cu-44.5 at.% Sn, denoted as $(Cu, Be)_6Sn_5$ in the binary phase diagram [21]. The other layer structure, near the substrate, was composed of Cu-25.2 at.% Sn, denoted as the $(Cu, Be)_3Sn$ phase. The atomic diffusion rate increased at the higher reaction temperature, which resulted in the rapid formation of the $(Cu, Be)_6Sn_5$ phase, and the Cu concentration could easily accumulate to form the Cu-rich $(Cu, Be)_3Sn$ phase. Therefore, the $(Cu, Be)_3Sn$ phase was observed under a reaction time of 0.5 h at 255 °C. The BEI micrograph of the reaction couple of SC/Alloy 25 reacted at a 255 °C reaction temperature for a 4 h reaction time is shown in Figure 4g. It was observed that the $(Cu, Be)_6Sn_5$ phase was divided into upper and lower layers with scallop-like and layer structures as the reaction time increased to 4 h. The upper was Cu-45.8 at.% Sn, and the lower was Cu-41.9 at.% Sn. From the size of the scallops, it was found that the $(Cu, Be)_6Sn_5$ phase ripening phenomenon was

significant. The BEI micrograph of the reaction couple of SC/Alloy 25 reacted at a 255 °C reaction temperature for a 10 h reaction time is shown in Figure 4h. As the reaction time was extended to 10 h, a clearer divided layer was observed, as shown in Figure 4h.

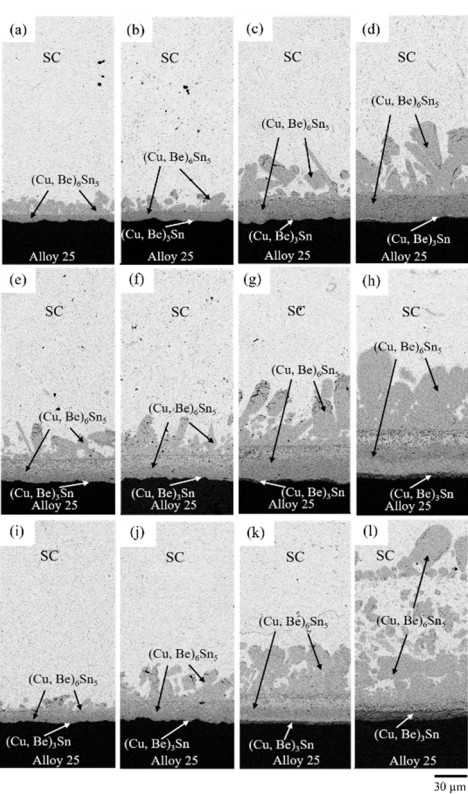

**Figure 4.** BEI micrographs of the SC/Alloy 25 reaction couples at 240 °C for (**a**) 0.5 h, (**b**) 1 h, (**c**) 4 h, and (**d**) 10 h; reacted at 255 °C for (**e**) 0.5 h, (**f**) 1 h, (**g**) 4 h, and (**h**) 10 h; and reacted at 270 °C for (**i**) 0.5 h, (**j**) 1 h, (**k**) 4 h, and (**l**) 10 h.

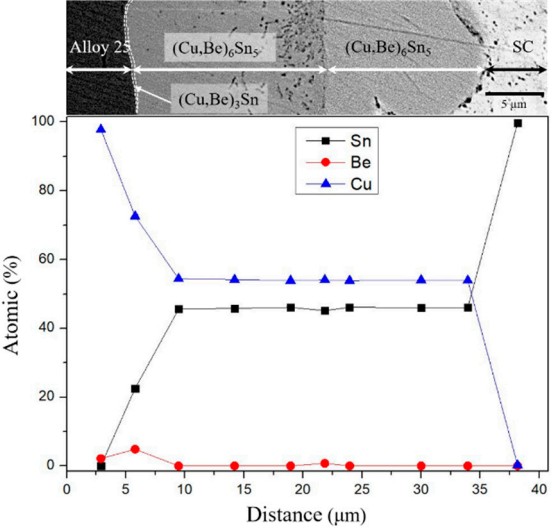

**Figure 5.** EPMA line scan and corresponding BEI micrograph of the SC/Alloy 25 reaction couple at a reaction temperature of 240 °C and reaction time of 1 h.

The BEI micrograph of the reaction couple of SC/Alloy 25 reacted at a 270 °C reaction temperature for a 0.5 h reaction time is shown in Figure 4i. A scallop-like structure was also observed on the solder side, as shown in Figure 4i. EDS analysis showed that the composition of SC/Alloy 25 reacted at a 270 °C reaction temperature for a 0.5 h reaction time was Cu-45.1 at.% Sn, denoted as the $(Cu, Be)_6Sn_5$ phase. Another thinner-layered structure was formed, determined as the $(Cu, Be)_3Sn$ phase [21]. Two types of $(Cu, Be)_6Sn_5$ formed as the reaction time of the SC/Alloy 25 reaction couple at 270 °C was increased to 4 h, as shown in Figure 4k. It can be speculated that at a prolonged reaction time, the phenomenon of grain ripening of the $(Cu, Be)_6Sn_5$ phase took place at the interface, during which the small grains merged into large grains. It has been reported that the grains are separated from the interface at a high system energy and diffuse into the solder to reduce the surface energy [30]. Yen et al. studied the interfacial reaction of SAC/Au/Ni/SUS304, which revealed that the $Cu_6Sn_5$ phase grains merged, became larger, and eventually left the substrate side at a prolonged reaction time [31]. The BEI micrograph of the reaction couple of SC/Alloy 25 reacted at a 270 °C reaction temperature for a 10 h reaction time is shown in Figure 4l. According to Figure 4l, the $(Cu, Be)_6Sn_5$ phase near the SC solder side became detached and proceeded to the SC solder as the reaction time increased to 10 h. However, the $(Cu, Be)_6Sn_5$ phase stratification disappeared. The thickness of the $(Cu, Be)_3Sn$ phase, on the other hand, increased significantly over time, resulting in a thicker $(Cu, Be)_3Sn$ phase.

A field emission dual-beam FIB microscope was used to observe the crystallinity of the $(Cu, Be)_6Sn_5$ phase. Figure 6a shows that the grain shape of the upper layer $(Cu, Be)_6Sn_5$ phase was thin, while the grain size of the lower layer $(Cu, Be)_6Sn_5$ phase was dense in the SC/Alloy 25 reaction couples reacted at 240 °C for 4 h. A schematic diagram of the AFM measurements is shown in Figure 6b. The middle layer of $(Cu, Be)_6Sn_5$ was slightly uneven, with a red mark. Figure 6c provides an AFM image of the SC/Alloy 25 reaction couples reacted at 240 °C for 4 h. The height difference was approximately 70 nm. The same results were obtained with SC/Alloy 25, reacted at 255 °C for 4 h, as shown in Figure 6d. The results showed that the crystal size at the upper layer $(Cu, Be)_6Sn_5$ was small, and the lower layer $(Cu, Be)_6Sn_5$ near $(Cu, Be)_3Sn$ was large. However, there were no micro-pits in the lower layer $(Cu, Be)_6Sn_5$ near $(Cu, Be)_3Sn$, as shown in Figure 6e, while the micro-pits existed on the upper layer, as in Figure 6f. Similar results were observed at 270 °C and 4 h, as shown in Figure 6g–i The results show that fewer diffusion channels were generated, which resulted in an increase in the Cu concentration and the easy formation of the $(Cu, Be)_3Sn$ phase. The $(Cu, Be)_6Sn_5$ phase grains in some micro-cavities were loosely arranged, while the light white band indicated the presence of Sn atoms. In addition, the grain was not completely formed in the middle. In Figure 5, EPMA results show that the composition was Cu-45.2 at.% Sn-0.7 at.% Be owing to some micro-cavities caused by the lattice distortion from the dissolution of Be into the $Cu_6Sn_5$ lattice. Moreover, the Be element appeared only in the $Cu_3Sn$ phase and the micro-cavities. It can be speculated that Be slowly diffused to the solder side along with Cu, and the Cu content in the $Cu_6Sn_5$ phases ranged from 52.2 at.% to 54.5 at.%. As the Alloy 25 reacted with SC, the Cu atoms in Alloy 25 were gradually consumed and diffused into the upper SC solder at the same time to form a scalloped $(Cu, Be)_6Sn_5$, which increased over time. The small grains were merged into large grains, and the grains ripened; the SC solder also diffused from the upper to the lower substrate. It can be speculated that as the grain moves closer to the Alloy 25, $(Cu, Be)_6Sn_5$ is formed with greater density.

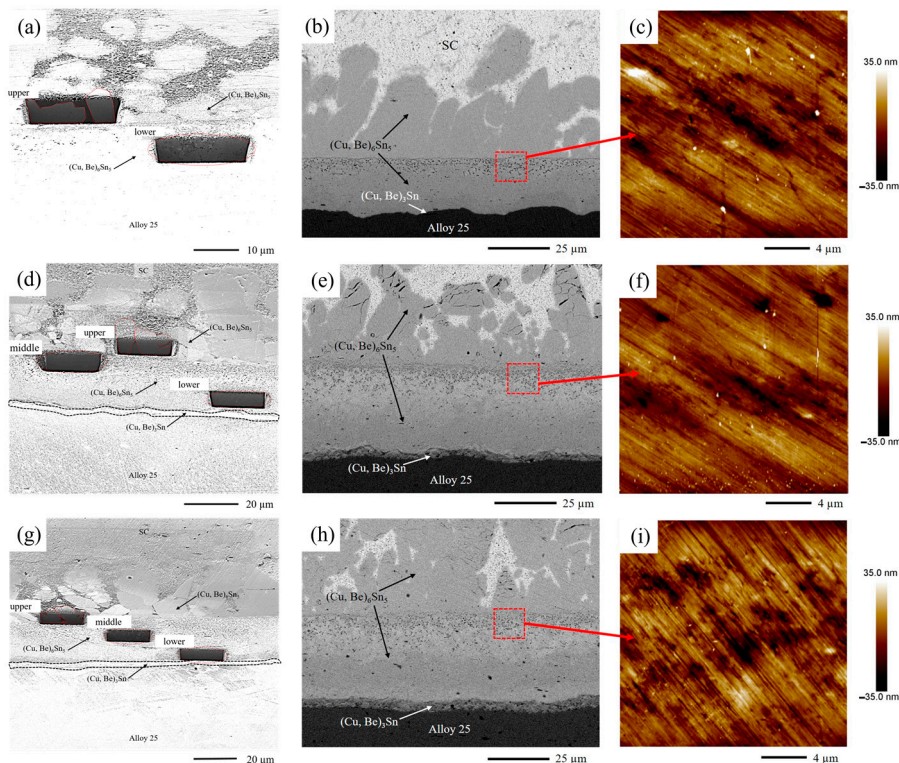

**Figure 6.** The SC/Alloy 25 reaction couples: (**a**) focused ion beam (FIB) images at 240 °C for 4 h, (**b**) schematic diagram of atomic force microscope (AFM) measurement at 240 °C for 4 h, (**c**) AFM image taken from (**b**), (**d**) FIB images at 255 °C for 4 h, (**e**) schematic diagram of AFM measurement at 255 °C for 4 h, (**f**) AFM image taken from (**e**), (**g**) FIB images at 270 °C for 4 h, (**h**) schematic diagram of AFM measurement at 270 °C for 4 h, (**i**) AFM image taken from (**h**).

### 3.3. SC/C1990 HP Reaction Couples

The BEI micrograph of the interface formed after the reaction between the SC solder and the C1990 HP substrate at a 240 °C reaction temperature for a 0.5 h reaction time is shown in Figure 7a. As shown in the figure, the IMC layer in the middle was characterized by a gray color. The composition of the IMC layer was Cu-45.3 at.% Sn, and it could be inferred as the $Cu_6Sn_5$ phase. The BEI micrographs of the reaction couple of SC/C1990 HP reacted at a 240 °C reaction temperature for reaction times of 0.5 h, 1 h, 2 h, 4 h, and 5 h are shown in Figure 7a–e. The temperature was fixed at 240 °C, and the time parameter was changed from 0.5 h to 5 h. The content of elements was measured using EDS, which depicted the presence of the $Cu_6Sn_5$ phase. According to Figure 7a–e, the $Cu_6Sn_5$ phase exhibited a significant scallop-like morphology [32,33] as the reaction time increased. Due to the larger wetting angle on the surface of the substrate Ti element, there was no adhesion between SC and C1990 HP, which was responsible for the easy IMC separation from the substrate. Hence, the prolonged reaction time resulted in the occurrence of smaller particles. The spheroids of $Cu_6Sn_5$ gradually left the substrate and peeled into the solder [16], along with the increase in the thickness of the IMC layer. The BEI micrographs of the reaction couple of SC/C1990HP reacted at a 255 °C reaction temperature for reaction times of 0.5 h, 1 h, 2 h, 4 h, and 5 h are shown in Figure 7f–j. According to Figure 7f–j, the $Cu_6Sn_5$ phase was present in the IMC layer formed by the reaction couple of SC/C1990 HP reacted at a 255 °C reaction temperature for different reaction times. The $Cu_6Sn_5$ phase of the IMC formed at 240 °C was consistent with the IMC phase found at 255 °C. However, the increase in the reaction time resulted in an increase in the thickness of the IMC layer. The BEI micrographs of the reaction couple of SC/C1990 HP reacted at 270 °C for reaction times of 0.5 h, 1 h, 2 h, 4 h, and 5 h are shown in Figure 7k–o. Furthermore, the results are consistent with those obtained at the temperatures of 240 °C and 255 °C. The EDS results showed that

the IMC layer was composed of the $Cu_6Sn_5$ phase. It could be speculated that the thickness of the IMC layer increased with the reaction time.

The BEI micrograph/EPMA of the reaction couple of SC/C1990HP, reacted at 255 °C for 2 h, is shown in Figure 8. The IMC, with a peeled layer, was composed of Cu-45.7 at.% Sn-2.2 at.% Ti, denoted as $Cu_6Sn_5$, with the copper content ranging from 51.11 at.% to 52.77 at.% [34], which is consistent with the literature.

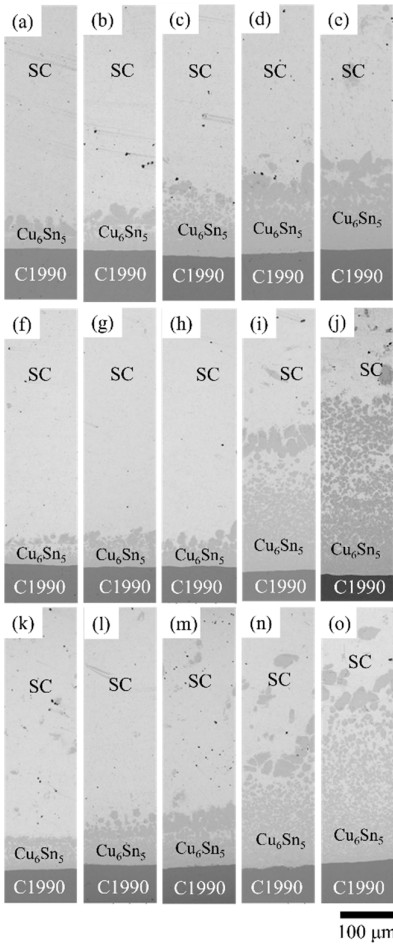

**Figure 7.** BEI micrographs of the SC/C1990 HP reaction couples at 240 °C for (**a**) 0.5 h, (**b**) 1 h, (**c**) 2 h, (**d**) 4 h, and (**e**) 5 h; reacted at 255 °C for (**f**) 0.5 h, (**g**) 1 h, (**h**) 2 h, (**i**) 4 h, and (**j**) 5 h; and reacted at 270 °C for (**k**) 0.5 h, (**l**) 1 h, (**m**) 2 h, (**n**) 4 h, and (**o**) 5 h.

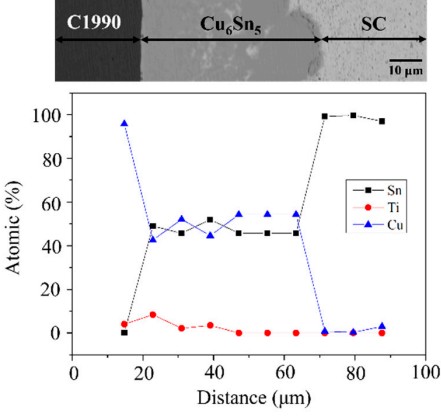

**Figure 8.** EPMA line scan and corresponding BEI micrograph of the SC/C1990 HP reaction couple at a reaction temperature of 255 °C and reaction time of 2 h.

*3.4. Kinetics in the Interfacial Reaction Couples*

The IMC thickness at the interface for the interfacial reaction meets the power law equation, $d = kt^n$ [35], where d is the thickness of the intermetallic phase (μm), $k$ is the growth rate constant (μm$^2$/s), t is the reaction time (s), and n is the time exponent of the IMC. Figure 9 shows the relationship between ln d and ln t at different reaction temperatures and reaction times for the SC/C194 reaction couple, where d is the thickness of the $Cu_6Sn_5$ phase, t is the reaction time, and n is the time exponent. A linear relationship can be obtained, and its slope reflects the n values, as listed in Table 3. The SC/C194 couple had n values of 1.05, 0.97, and 0.96 at three different reaction temperatures of 240 °C (1–10 h reaction time), 255 °C (1–10 h reaction time), and 270 °C (0.5–10 h reaction time), respectively. These n values are close to 1, indicating that the IMC growth mechanism is determined by the interfacial reaction [35]. Meanwhile, n values close to 0.5 were found at 240 °C (0.5–1 h) and 255 °C (0.5–1 h). This indicates that the IMC growth mechanism is controlled by the interfacial reaction in general, which facilitates rapid atom diffusion to the interface. Moreover, the formation of the IMC determines the overall reaction rate. The $(Cu, Fe)_6Sn_5$ phase was dispersed onto the solder side. In contrast, the denser $(Cu, Fe)_6Sn_5$ phase was not generated because the trace Fe atoms played the role of heterogeneous nucleation sites in the substrate. According to Figure 1b,e,h, in the case of SC/C194, a large number of IMCs dispersed into the solder side at a 2 h reaction time. To maintain the stable state of the interface, new IMCs were generated and dispersed into the solder immediately. It can be speculated that the generation rate of IMCs in the SC/C194 couple was completely determined by the rate of reaction generation.

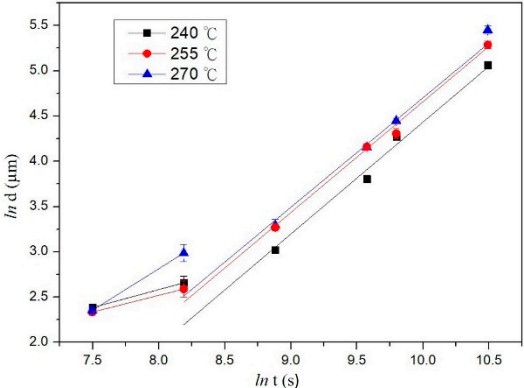

**Figure 9.** The plot of ln t vs. ln d of $Cu_6Sn_5$ phase at different reaction temperatures in the SC/C194 reaction couple.

**Table 3.** The n values of the $Cu_6Sn_5$ phase at different reaction temperatures in the SC/C194 reaction couple.

| Temperature | Time | n |
|---|---|---|
| 240 °C | 0.5~1 h | 0.46 |
| 240 °C | 1~10 h | 1.05 |
| 255 °C | 0.5~1 h | 0.47 |
| 255 °C | 1~10 h | 1.04 |
| 270 °C | 0.5~10 h | 0.96 |

Figure 10 shows the relationship between ln d and ln t for the SC/Alloy 25 reaction couple at different reaction temperatures and different reaction times. It was observed that the thickness of the $(Cu, Be)_6Sn_5$ phase increased with temperature and time. In Table 4, the n value ranges from 0.28 to 0.34 for the three different reaction temperatures of 240 °C, 255 °C, and 270 °C. These three n values are close to 0.33, which indicates that the IMC growth mechanism was determined by grain boundary diffusion [35]. It was believed

that Be atoms dissolved in the $Cu_6Sn_5$ phase to form the $(Cu, Be)_6Sn_5$ phase. To create crystal defects, the grain boundary of the $Cu_6Sn_5$ phase forms a short-range channel for diffusion, which promotes diffusion. When the diffusion time of atoms along the grain boundary was much longer than the time of diffusion to the interface and crossing the interface, the reaction was limited via controlling the grain boundary diffusion [36]. The Be atoms in the SC/Alloy 25 system were solid and dissolved into the $Cu_6Sn_5$ phase, which may have resulted in lattice distortion for the formation of short-range channels at the grain boundaries of the $Cu_6Sn_5$ phase for diffusion. It can be speculated that the generation rate of IMCs was controlled by grain boundary diffusion.

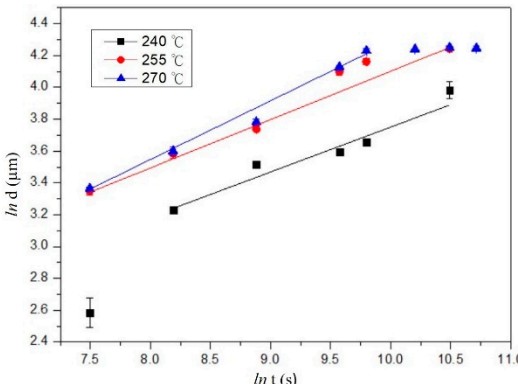

**Figure 10.** The plot of ln t vs. ln d of $Cu_6Sn_5$ phase at different reaction temperatures in the SC/Alloy 25 reaction couple.

**Table 4.** The n values of $Cu_6Sn_5$ phase at different reaction temperatures in the SC/Alloy25 reaction couple.

| Temperature | Time | n |
|---|---|---|
| 240 °C | 1~10 h | 0.28 |
| 255 °C | 0.5~10 h | 0.30 |
| 270 °C | 0.5~5 h | 0.34 |

The growth rate constant ($\mu m^2$/s or $m^2$/s) of an IMC is calculated as $d = k\ (t)^{1/2}$ [37]. After the value of *k* is obtained, the equation $\ln k = \ln k_0\ [(-Q)/R] \times (1/T)$ [37] can be used to find Q, where $k_0$ is the pre-exponential factor ($\mu m^2$/s), Q is the activation energy of the reaction (kJ/mole), R is the ideal gas constant (8.314 J/K mole), and T is the reaction temperature (Kelvin (K)).

Figure 11 illustrates the relationship between the thickness of the $Cu_6Sn_5$ phase and the square root of the reaction time for the SC/C1990 HP couples at different temperatures. It was observed that the thickness of the $Cu_6Sn_5$ phase increased with the increase in temperature and time. The average thickness of the scallop-shaped $Cu_6Sn_5$ layer formed via the reaction of SC/Cu at 280 °C in the previous literature was 17.8 μm [38]. However, the thickness of the $Cu_6Sn_5$ exfoliation in this experiment was much greater. The results showed that the thickness of the $Cu_6Sn_5$ phase was proportional to the time. Meanwhile, IMC growth obeyed the parabolic law. The calculated slope for the three temperatures (240 °C, 255 °C, 270 °C) was approximately 0.5, indicating that the interface reaction between the SC solder and the C1990 HP substrate was diffusion-controlled. The diffusion control leads to the rapid growth of the reaction into $Cu_6Sn_5$, and it is quickly stripped into the solder.

The ln *k* against 1/T (Figure 12) was plotted to obtain the slope-(Q/R), which was used to calculate the activation energy (Q) of the SC/C1990 HP couple. The calculated growth coefficients and activation energies of IMCs produced by SC/C1990HP in the present study were compared with the literature, as shown in Table 5. The comparison study showed that the IMCs formed in the present study had higher activation energy, which indicates

that the formation and growth of IMC are remarkably difficult. Conversely, low activation energy indicates that the formation and growth of IMCs are easier [39]. In the case of the C1990 HP substrate, the $Cu_6Sn_5$ phase produced by SC/C1990 HP showed a longer peeling distance in the direction of the solder than SAC/C1990 HP. As a result, the activation energies were too high compared to Sn/Alloy 25 [15], SAC/Alloy 25 [15], SAC/C1990 HP [16], and SAC/C194 [40]. It included the mechanism of IMC growth and peeling, so that the entire k value measured included the growth and peeling of the phase. The peeling was caused by the $Cu_6Sn_5$ phase drifting into the molten solder. It was observed that the drift rate was much slower than the growth rate of the phase. The drift situation promoted an increase in the activation energy, similar to the addition of $TiO_2$ nanoparticles to an SAC solder and Cu substrate in the literature [41]. Previous results show that the addition of $TiO_2$ may inhibit the growth of the whole IMC layer in the solid state. With the increase in $TiO_2$ content, the growth rate of the whole IMC layer decreases, and the activation energy increases, which is consistent with the present study.

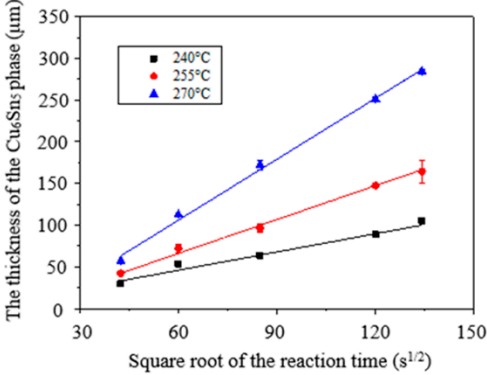

**Figure 11.** The thickness of the $Cu_6Sn_5$ phase (μm) relative to the square root of reaction time ($s^{1/2}$) at different reaction temperatures in the SC/C1990 HP.

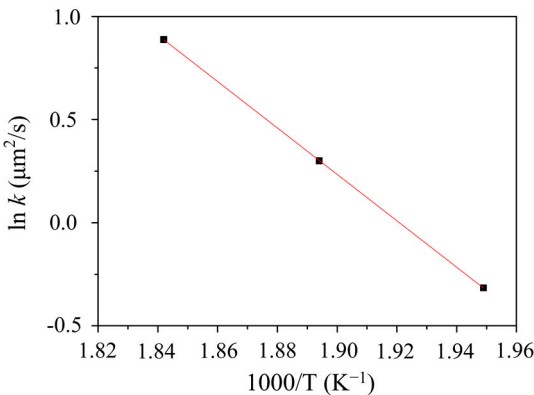

**Figure 12.** The Arrhenius plots of the growth rate constant of the $Cu_6Sn_5$ phase (ln k) vs. 1000/T in the SC/C1990 HP.

**Table 5.** Growth rate constant (*k*) and activation energy (*Q*) of the IMC for the SC/C194, SC/Alloy 25, and SC/C1990HP couples reacted at various temperatures.

| System | Growth Rate Constant *k* ($\times 10^{14}$ m²/s) | | | Q (kJ/mole) | Reference |
|---|---|---|---|---|---|
| | 240 °C | 255 °C | 270 °C | | |
| SC/C1990 HP | 53.0 | 180.0 | 590.0 | 186.0 | This Study |
| Sn/Alloy 25 | 273.8 | 531 | 286.8 | 27.3 | [15] |
| SAC/Alloy 25 | 162.8 | 351.3 | 342.9 | 58.0 | [15] |
| SAC/C1990 HP | 73.1 | 80.8 | 123.9 | 40.6 | [16] |
| SAC/C194 | - | - | - | 45.2 | [40] |

## 4. Conclusions

This study investigated the interfacial reaction between the SC solder and Cu-based alloys (C194, Alloy 25, and C1990 HP) at different reaction temperatures ranging from 240 °C to 270 °C and different reaction times. Moreover, the liquid–solid reaction and the growth kinetics of IMCs were determined. Based on the experimental results and discussion above, the following conclusions can be drawn:

1.  The Fe atoms from the C194 substrate provide good nucleation sites for the IMC reaction that results in the formation of the $(Cu, Fe)_6Sn_5$ phase, which cannot be dense and thick.
2.  The Be atoms from Alloy 25 are the solids dissolved into the $(Cu, Be)_3Sn$ and $(Cu, Be)_6Sn_5$ phases, resulting in lattice distortion with unevenness.
3.  The Ti element in the C1990 HP substrate facilitates heterogeneous nucleation, resulting in no formation of other IMCs, which is responsible for the longer diffusion distance in the case of $Cu_6Sn_5$.
4.  The IMC growth mechanism of each system has different types, which are reaction control, grain boundary diffusion control, and diffusion control for the SC/C194, SC/Alloy 25, and SC/C1990 HP systems, respectively.

**Author Contributions:** Conceptualization, Y.-W.Y.; Formal analysis, A.D.L.; Investigation, T.-Y.T. and T.-H.C.; Methodology, Y.-W.Y.; Project administration, Y.-W.Y.; Supervision, Y.-W.Y.; Writing—original draft, Y.-C.C.; Writing—review and editing, A.D.L. All authors have read and agreed to the published version of the manuscript.

**Funding:** The authors acknowledge financial support from the National Science and Technology Council, Taiwan, R.O.C. (Grant No. MOST 110-2221-E-011-049 and MOST 111-221-E-011-110-MY3) and the Applied Research Center for Thin-Film Metallic Glass from the Featured Areas Research Center Program within the framework of the Higher Education Sprout Project of the Ministry of Education (MOE) in Taiwan.

**Acknowledgments:** The authors are also grateful for the assistance of S. C. Laiw, who works at the National Taiwan University of Science and Technology, for the SEM-EDS operation.

**Conflicts of Interest:** The authors declare no conflict of interest.

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
