# Peer review of "Investigation of the Sn-0.7 wt.% Cu Solder Reacting with C194, Alloy 25, and C1990 HP Substrates"

_metals, doi:10.3390/met13010012_

Round 1
Reviewer 1 Report
In this work, the authors studied the interaction between a molten Sn-0.7wt.% Cu solder and three Cu-based alloys. The work is interesting and novel. It provides some fundamental kinetic data for the studied liquid-solid interaction. It is publishable subject to revision.
1.It is necessary to specify the chemical composition of the studied alloys (C194, Alloy 25 and C1990 HP) since readers may not be familiar with the commercial nomenclature. The chemical composition should be given in the form of a table. The table should be presented in the materials and methods section. Any trace elements should be specified.
2.Table 1 lists the chemical compositions of the IMCs formed at the solder-substrate interface. The table should be simplified since identical IMCs were found at each interface regardless of temperature. Also, there are misprints. Compounds (Cu,Fe)3Sn5 and (Cu,Be)3Sn5 should be (Cu,Fe)3Sn and (Cu,Be)3Sn since only Cu6Sn5 and Cu3Sn compounds are known in the Cu-Sn binary system at temperatures below 300 °C.
3.The kinetic data are compared in Fig. 9. The thickness of the IMC formed on the surface of the Cu-Be alloy is considerably smaller compared to the Cu-Ti and Cu-Fe alloys. Be has a very small atomic radius compared to those of Fe and Ti. Therefore, the diffusion of this element in the molten solder might have been fundamentally different. Have you checked the diffusion coefficients of Fe, Ti, and Be in molten Sn or Sn-0.7Cu in literature? Are they comparable or fundamentally different?
4.Beryllium may create a large lattice distortion of the (Cu,Be)5Sn6 compound because of its small atomic radius. Have you measured some XRDs of the materials? If so, present them in the manuscript.
5.The authors fit all data with a parabolic rate law (Eq 1, line 332). It is correct for the Cu-Fe and Cu-Ti alloys. However, the thickness of the layer found on the Cu-Fe substrate follows a linear rate law (Fig. 9 a). Therefore, the interaction is controlled by the surface reaction. The linear rate constants should be calculated and presented in Table 2 for this system. Furthermore, the activation energy should be recalculated for the SC-C194 system based on linear rate constants.
6.Conclusions should be given point-by-point.
Author Response
Answers to Reviewer 1's comments:
A-1: We are very grateful for Reviewer 1’s valuable advice. By following Reviewer 1’s advice, the new table (Table 1) of the chemical composition of C194, Alloy 25, and C1990 HP has been added in the "Materials and Methods" section.
A-2: We are very grateful for Reviewer 1’s valuable advice. We have revised the misprints in Table 1 (updated become Table 2). In addition, we have added an explanation of the identical the (Cu, Be)6Sn5 phase with a red mark in line 163-164.
A-3: We are very grateful for Reviewer 1’s valuable advice. There are several literatures that discuss the diffusion coefficients of Fe and Ti in Sn. According to Sarafianos, with a low concentration of Fe in the Fe-Sn system, the diffusion coefficients are 2.16 × 10-11 cm2/s [1]. Meanwhile, according to Wang et. al., the diffusion coefficient of Ti in Sn is assumed to be the same as the self-diffusivity of Sn in Ti, which is 6.9 × 10-1 cm2/s. From the literatures, it was found that the Ti and Fe elements in Sn have different diffusion coefficients. However, there is no literature related to diffusion coefficients on Be in molten Sn.
[1] Sarafianos, N., An analytical method of calculating variable diffusion coefficients. J. Mater. Sci. 1986, 21, 2283-2288.
[2] Wang, J. L.; Liu, L. B.; Tuo, B. Y.; Bai, W. M.; Wang, X., Li, X.; Hu, X. P. Computational study of mobilities and diffusion in Ti-Sn alloy. J Phase Equilibria Diffus. 2015, 36, 248-253.
A-4: We are very grateful for Reviewer 1’s evaluation of our manuscript. However, we didn't measure some XRD tests in this study. On your valuable suggestion, we will do the XRD tests in the next research.
A-5: We are very grateful for Reviewer 1’s evaluation of our manuscript. We have updated Table 2 (update: now it is Table 5) for the Q value according to the parabolic law, which is only on the SC/C1990 HP system. Meanwhile, for the SC/194 and SC/Alloy 25 systems, we show the ln t vs ln d graphs (Figure 9 and Figure 10) and Tables of n values (Table 3 and Table 4).
A-6: We are very grateful for Reviewer 1’s valuable advice. The conclusions have been given point by point in the manuscript.
Reviewer 2 Report
Dear Authors,
It was really pleasure to read your paper. I am under the effect of the work and yours knowledge and experience in the topic of soldering.
In my opinion the valuable question is described the reaction in interface area in evaluated alloys. I agree with authors that it can improve the use of this alloys in industry.
I believe this topic is original. Preparing for the review I found only a few papers in which this type of alloys were considered in the soldering process. In reviewed paper it was done in most complex way. I think that there are complex researches. The authors tried to explain the reactions and the diffusion mechanism in the interface area. In some publications you can find only description of the structure and morphology of IMC without analysis of their origin.
Some minor suggestions:
1)In my opinion the conclusions are supported by results and analysis. Maybe it should be listed in points.
2)Regarding the methodology, I think that they should add the tables of used materials chemical composition.
3)I have found one error. In the line 47-48 you enter the dimensions of the sample. I think that the mm would be enough instead of mm3
Best regards
Author Response
A-1: We are very grateful for Reviewer 2’s valuable advice. The conclusions have been given point by point in the manuscript.
A-2: We are very grateful for Reviewer 2’s valuable advice. By following the reviewer’s advice, the new table (Table 1) of the chemical composition of the C194, Alloy 25, and C1990 HP) has been added in the "Materials and Methods" section.
A-3: We are very grateful for Reviewer 2’s valuable advice. We have revised the error of the unit in the manuscript.
Reviewer 3 Report
The submitted manuscript entitled “Investigation on the Sn-0.7 wt.% Cu Solder Reacting with C194, Alloy 25, and C1990 HP Substrates” deals with the interfacial reactions in Sn-0.7 wt.% Cu (SC) lead-free solder reacting with Cu-3.3 wt.% Fe (C194), Cu-2.0 wt.% Be (Alloy 25), and Cu-3.3 wt.% Ti (C1990HP). The intermetallic compounds formed at the interface were investigated in many aspects: diffusion mechanism, growth rate, composition and the morphology. The manuscript is interesting, during its review, the following issues arose.
The experiment consisted of a very simple way. The paper lacks scientific originality.
The review of the literature contained in the ‘Introduction’ section is very superficial.
For a better presentation of the results in the printed version of the article, it is suggested to enlarge figures 1, 2, 4 and 7.
Lines 152-153 and 173-174: the sentences are the same, however different figures have been quoted.
Caption of the figure 5: "FIB", "AFM". Although these are well-known acronyms, they should be explained at first use.
Sentence in lines 220-221 should be moved to the "Materials and Methods" section.
The font size in figures 9 and 10 should be enlarged.
Please check the equation (1).
Sources of the Eqs. (1) and (2) should be cited.
The article lacks discussion of the results in comparison to other articles as well as implications for academic and commercial matters.
Author Response
A-1: We are very grateful for Reviewer 3’s valuable advice. In the "Introduction" section, we add information that C194, Alloy 25, and C1990 HP substrates have the potential to replace Cu conductors as lead-frame materials. We also add several references that have conducted research on the three substrates. Research on the interfacial reaction between Sn with 0.7 wt.% Cu addition as a solder by reacting C194 substrate still needs to be investigated. The Liquid/solid reaction between Sn-07 wt.% Cu and C194 is important to reveal the potential for new lead-frame innovations to replace Cu. In addition, we added some information in the "Materials and Methods" section including the composition of each substrate.
A-2: We are very grateful for Reviewer 3’s valuable advice. We have enlarged Figures 1, 2, 4, and 7.
A-3: We are very grateful for Reviewer 3’s valuable advice. We have revised the temperature in lines 173-174 (updated lines 184-185). It should be 255℃, not 240℃. In addition, we changed the sentence so that it is not the same as before.
A-4: We are very grateful for Reviewer 3’s valuable advice. We have explained the acronyms of "FIB" and "AFM" in Figure 5.
A-5: We are very grateful for Reviewer 3’s valuable advice. The AFM and FIB instruments used have been moved to the "Materials and Methods" section.
A-6: We are very grateful for Reviewer 3’s valuable advice. Based on the suggestion of other reviewers, there are revisions from Figures 9 and 10. In addition, we have enlarged the font size of Figures 9 and 10. We change the graph in Figure 11 to show the IMC growth obeyed the parabolic law and support the calculation of the activation energy in the SC/C1990 HP couple.
A-7: We are very grateful for Reviewer 3’s valuable advice. We have checked Equation (1) and have provided citations from Equations (1) and (2). We write these two equations in the text in the revised manuscript.
A-8: We are very grateful for Reviewer 3’s valuable advice. The references have been added to support the results and discussion. The updated information and references have been marked in red fonts in the “Results” section. Here is the list of references that have been added in the revised manuscript.
[13] Ong, C.G.; Lau, K.T.; Zaimi, M.; Afiq, M.; Queck, K.P. Effect of electroless Ni-P thickness on EFTECH 64-Ni, EFTECH 64-Cu and C194-Ni bump. In Proceedings of the 11th International Microsystems, Packaging, Assembly and Circuits Technology Conference (IMPACT), Taiwan, 26-28 October 2016.
[14] Liu, T.; Ding, D.; Hu, Y.; Gong, Y. Effect of interfacial reaction on tin whisker formation of Sn/Ni films deposited on copper lead-frame. In Proceedings of the 15th International Conference on Electronic Packaging Technology, China, 12-15 August 2014.
[35] Yin, Z.; Lin, M.; Li, Q.; Wu, Z. Effect of doping Ni nanoparticles on microstructure evolution and shear behavior of Sn–3.0Ag–0.5Cu(SAC305)/Cu–2.0Be solder joints during reflowing. J. Mater. Sci.: Mater. Electron. 2020, 31, 4905-4914.
[36] Shen, J.; Chan, Y.C.; Liu, S.Y. Growth mechanism of Ni3Sn4 in a Sn/Ni liquid/solid interfacial reaction. Acta Mater. 2009, 57, 5196-5206.
Round 2
Reviewer 1 Report
Authors answered most of my comments. The paper is acceptable for publication.